# Cyclosporine A Protects Retinal Explants against Hypoxia

**DOI:** 10.3390/ijms221910196

**Published:** 2021-09-22

**Authors:** Sven Schnichels, Maximilian Schultheiss, Patricia Klemm, Matthias Blak, Thoralf Herrmann, Marion Melchinger, Karl-Ulrich Bartz-Schmidt, Marina Löscher, Günther Zeck, Martin Stehphan Spitzer, José Hurst

**Affiliations:** 1Centre for Ophthalmology Tübingen, University Eye Hospital Tübingen, 72076 Tübingen, Germany; sven.schnichels@googlemail.com (S.S.); Patricia.Klemm@gmx.de (P.K.); MatthiasBlak@gmail.com (M.B.); m.melchinger@web.de (M.M.); U.Bartz-Schmidt@uni-tuebingen.de (K.-U.B.-S.); ml.nanoidrops@gmail.com (M.L.); spitzko@googlemail.com (M.S.S.); 2Clinic for Ophthalmology, University Medical Center Hamburg-Eppendorf (UKE), 20246 Hamburg, Germany; maximilianschultheiss@gmail.com; 3Department of Ophthalmology, Klinikum Stuttgart, 70174 Stuttgart, Germany; 4NMI Natural and Medical Sciences Institute, University of Tübingen, 72770 Reutlingen, Germany; therrmann@nmi.de; 5Institute of Electrodynamics, Microwave and Circuit Engineering, TU Wien, 1040 Vienna, Austria; guenther.zeck@tuwien.ac.at

**Keywords:** hypoxia, ischemia, cyclosporine A, retinal ganglion cells, neuroprotection

## Abstract

The retina is a complex neurological tissue and is extremely sensitive to an insufficient supply of oxygen. Hypoxia plays a major role in several retinal diseases, and often results in the loss of cells that are essential for vision. Cyclosporine A (CsA) is a widely used immunosuppressive drug. Furthermore, treatment with CsA has neuroprotective effects in several neurologic disorders. No data are currently available on the tolerated concentration of CsA when applied to the retina. To reveal the most effective dose, retinal explants from rat eyes were exposed to different CsA concentrations (1–9 µg/mL). Immunohistochemistry with brain-specific homeobox/POU domain protein 3a (Brn3a) and TUNEL staining was performed to determine the percentage of total and apoptotic retinal ganglion cells (RGCs), as well as the responses of micro- and macroglial cells. Furthermore, optical coherence tomography (OCT) scans were performed to measure the changes in retinal thickness, and recordings with multielectrode array (MEA) were performed to evaluate spontaneous RGC spiking. To examine the neuroprotective effects, retinas were subjected to a hypoxic insult by placing them in a nitrogen-streamed hypoxic chamber prior to CsA treatment. In the biocompatibility tests, the different CsA concentrations had no negative effect on RGCs and microglia. Neuroprotective effects after a hypoxic insult on RGCs was demonstrated at a concentration of 9 µg/mL CsA. CsA counteracted the hypoxia-induced loss of RGCs, reduced the percentage of TUNEL^+^ RGCs, and prevented a decrease in retinal thickness. Taken together, the results of this study suggest that CsA can effectively protect RGCs from hypoxia, and the administered concentrations were well tolerated. Further in vivo studies are needed to determine whether local CsA treatment may be a suitable option for hypoxic retinal diseases.

## 1. Introduction

The neuroretina is a highly complex nervous tissue that is essential for vision. It is composed of various cell types, including photoreceptors, microglia, as well as bipolar, horizontal, retinal ganglion, Müller, and amacrine cells. Together, they form a complex three-dimensional cellular network that represents a functional unit. Retinal hypoxia is a common cause of several diseases leading to visual impairment or even blindness. At the cellular level, retinal hypoxia activates a self-reinforcing signaling cascade that results in neuronal depolarization, calcium influx, and oxidative stress [1].

Diseases in which hypoxia plays a key role include glaucoma, diabetic retinopathy, age-related macular degeneration (AMD), and central retinal artery occlusion (CRAO) [2,3,4,5,6]. 

After cataract, the second leading cause of irreversible blindness worldwide is glaucoma [7]. The underlying pathomechanisms of glaucoma are not fully understood, and the current treatments aim only to slow disease progression. However, there are numerous causes of glaucoma that ultimately lead to retinal ganglion cell (RGC) death [8,9,10]. Both hypoxia and oxidative stress are known molecular factors involved in glaucoma [11,12,13]. 

Cyclosporine A (CsA) is a widely used immunosuppressive drug. Additionally, CsA has several anti-apoptotic properties and initiates neuroprotective processes if administered after ischemic stroke or traumatic brain injury [14,15,16,17,18,19]. The main mechanism of its function seems to be the inhibition of the mitochondrial permeability transition pore (mPTP) and the calcium-dependent phosphatase calcineurin [18,20,21]. In different neuronal cell-types, CsA was demonstrated to be protective against glutamate-induced neuronal cell damage [22,23,24]. Furthermore, Kim et al. reported that CsA was neuroprotective in retinal ischemia induced by elevated intraocular pressure. CsA was administered in 4-month-old C57BL/6 mice once a day with a concentration of 5 mg/kg. Both CsA and vehicle control were injected once by intraperitoneal injection at 24 h before and at the time of initial IOP elevation [25]. 

In prior studies, we used the RGC-5 cell line and administered CsA directly to the RGC-5 cell culture [26]. In those experiments, 9 µg/mL exhibited the best neuroprotective results in cells stressed with 10 mM glutamate for 24 h. Glutamate induced a significant decrease in overall cell viability. The supplementation of 9 μg/mL CsA ensured the survival of RGC-5 cells by preventing the induction of apoptosis.

To confirm our prior results—especially given the concerns regarding the origin of the RGC-5 cell line—, here, we used our previously published retinal hypoxia rat organ culture model [27,28,29,30,31]. The biocompatibility of 1, 3, 6, and 9 µg/mL CsA and its neuroprotective potential were evaluated. In a clinical situation, local treatment in the form of an intravitreal injection would be desirable. Intravitreal injections are well established in ophthalmology, and the frequency of injections depends on the local half-life [32]. To avoid toxic effects on the retina, the tolerable concentration of CsA needs to be evaluated on a cellular basis. In this experimental setup, CsA was administered directly to the retina in the medium, which is the closest ex vivo stimulation of an intravitreal injection. At some concentrations, a rescue effect on hypoxia-induced damage was observed. These findings further support CsA as a potential therapeutic agent for diseases affecting RGC-like glaucoma or central artery occlusion, because CsA could possibly prevent RGC loss and mitigate a harmful immune system response.

## 2. Results

### 2.1. Biocompatibility of Cyclosporine A

To evaluate the biocompatibility of CsA, rat neuroretinal explants were treated with different CsA concentrations ranging from 1 to 9 µg/mL. After 48 h, the number of RGCs and the rate of TUNEL^+^ RGCs was determined (Figure 1A). Quantification of the total amount of cells (DAPI^+^ cells) and the number of brain-specific homeobox/POU domain protein 3A (Brn3a^+^) positive RGCs in the retinal ganglion cell layer (GCL) was performed. The amount of Brn3a^+^ cells was correlated with the total number of cells in the GCL. In detail, quantification of RGCs after 48 h of incubation revealed 19.1% Brn3a^+^ cells in the GCL in the control group. In the CsA-treated groups, we detected no significant change in the proportion of RGCs (1 µg/mL: 19.8%; 3 µg/mL: 20.8%; 6 µg/mL: 19.7%; 9 µg/mL: 17.8% of Brn3a^+^ cells) (Figure 1B). Analysis of the percentage of TUNEL^+^ RGCs did not reveal significant increase in any of the CsA concentrations at any of the two investigated time points compared to the controls, in the 6 µg/mL treated group no TUNEL^+^ cells were found (Figure 1C). Taken together, it can be assumed that none of the tested CsA concentrations influenced cellular viability, therefore displaying a good biocompatibility of these concentrations. The immunosuppressive drug CsA may affect inflammation and the wound healing process. It is also possible that there could be indirect effects of CsA on glial cells via other cell types. Therefore, the influence of CsA treatment on macro- and microglial cells was also investigated by staining the markers glial fibrillary acidic protein (GFAP) and cluster of differentiation molecule 11B (CD11b) (Figure 2). To evaluate gliosis, the percentage of GFAP^+^ or CD11b^+^ fluorescence area was measured, and the change compared with the control group was assessed. Compared to the controls, only 6 µg/mL CsA induced a significant increase in the GFAP-stained area after 24 h (control: 4.2%; 6 µg/mL CsA: 6.62%; *p* < 0.01) (Figure 2A,C). However, at a concentration of 9 μg/mL, the value dropped again to 3.49%. The examination of microglia gave no indication of possible gliosis due to CsA (Figure 2B,D).

### 2.2. Hypoxia and Cyclosporine A Treatment Induced a Change in Retinal Thickness

Since cell death results in a decrease of retinal thickness, the width of the retina was determined via optical coherence tomography (OCT) after 24 h and 48 h of cultivation [12]. Because the two higher concentrations did not lead to a negative effect, the further studies were only conducted with 6 µg/mL and 9 µg/mL of CsA. Measurements were taken after 4, 24, and 48 h. The 4 h time point served as a control and the thickness of the other two time points was calculated as a relative change compared to the control, which was set as 100%. Treatment with 6 and 9 µg/mL CsA alone did not alter retinal thickness after either 24 h or 48 h (Figure 3). Hypoxia induced a significant reduction in retinal thickness after 24 h (93.62%; *p* < 0.05) compared to the controls. The reduction was more pronounced after 48 h (83.06%; *p* < 0.001). The addition of CsA counteracted the effect of retinal thinning in a dose-dependent manner (Figure 3). Compared to the hypoxia group, treatment with 6 µg/mL CSA prevented thinning after 24 h (100.8%; *p* < 0.05) but not after 48 h (91.7%; *p* > 0.05). At a concentration of 9 µg/mL, CsA protected the retinal explants from thinning at both time points (24 h: 104.6%, *p* < 0.001; 48 h: 98.2%, *p* < 0.001) and ameliorated the hypoxic effect completely (Figure 3B). 

### 2.3. Cyclosporine A Increased Cell Survival after Hypoxic Insult

After preventing hypoxia-induced retinal thinning with CsA, the survival of RGCs was investigated. Again, in order to obtain a better comparison, the amount of RGCs in controls was set as 100%. Hypoxic treatment for 75 min significantly reduced the percentage of RGCs (73%; *p* < 0.05) compared to controls (Figure 4A,C). The addition of 6 µg/mL CsA did not promote RGC survival compared to the hypoxia group (52.3%; *p* < 0.01 related to control). Interestingly, the addition of 9 µg/mL CsA resulted in a complete and highly significant rescue effect (120%; *p* < 0.001 with respect to hypoxic treatment).

In line with these results, the percentage of TUNEL^+^ RGCs was significantly increased after hypoxic treatment compared to the controls (control group: 1%; hypoxia group: 15%; *p* < 0.01). Both CsA concentrations attenuated the hypoxic effect completely (6 µg/mL CsA: 1.8%; 9 µg/mL CsA: 2% TUNEL^+^ cells; *p* < 0.05) and significantly reduced the number of TUNEL+ RGCs compared to the hypoxia group (Figure 4C). 

These encouraging results seen in immunohistochemistry could not be confirmed by qRT-PCR on the expression of the RGC marker beta-3-tubulin (TUBB3). Hypoxia led to a significant 3-fold decrease in mRNA expression (*p* < 0.001), which could not be prevented by any CsA treatment (Figure 4D).

### 2.4. Glial Cell Response

To investigate the response of glial cells and to verify the potential positive effects of CsA treatment on glial cells under hypoxia, we performed staining of the eye sections with antibodies against GFAP and CD11b (Figure 5A,B). Areas of DAPI, GFAP^+^, and CD11b^+^ signals were evaluated with ImageJ software and the percentage of the GFAP^+^ or CD11b^+^ area compared to the total area (DAPI) was calculated. After hypoxic treatment, the GFAP^+^ area increased significantly (*p* < 0.01) to 6.19% compared to the controls (4.26%). The 6 μg/mL CsA did not decrease the GFAP^+^ area but 9 μg/mL CsA ameliorated the GFAP^+^ area (6 μg/mL CsA: 7.61%, *p* < 0.001 with respect to the control condition and *p* < 0.05 with respect to hypoxic treatment; 9 μg/mL CsA: 5.03%, *p* > 0.05%) (Figure 5C). 

In contrast to GFAP, the expression of CD11b was not significantly altered after hypoxic treatment. Indeed, hypoxia for 75 min led to a reduction in microglia after 48 h compared to controls (control: 2.07%; hypoxia: 1.82%) (Figure 5D). Interestingly, both CsA concentrations increased the percentage of microglia (6 μg/mL CsA: 2.97%, *p* < 0.05 with respect to hypoxic treatment; 9 μg/mL CsA 2.8%, *p* > 0.05). 

### 2.5. Cyclosporine A Attenuated the Loss of Spontaneous RGC Activity after Retinal Hypoxia

MEA recordings with 6 or 9 µg/mL CsA were performed after 48 h. The spontaneous RGC activities of the controls were set to 100%. The supplementation of 6 or 9 µg/mL CsA to the medium did not negatively affect spontaneous RGC activity, whereas no spontaneous RGC spiking was detectable in the hypoxia group (Figure 6). The loss of spontaneous RGC activity induced by hypoxia could not be reversed with treatment of 6 µg/mL CsA, no RGC activity was detectable in this group either. However, the loss of function after retinal hypoxia was counteracted but not reversed by 9 µg/mL CsA. A 30% activity of RGC was detected (*p* < 0.01).

## 3. Discussion

The potential neuroprotective effects of cyclosporine A (CsA) have been demonstrated not only in hypoglycemic events, but also in ischemic insults in the brain [33,34]. The potential neuroprotective role of CsA in ischemia was first published by Shiga and colleagues. Both the edema formation and infarct size after focal cerebral hypoxia were reduced by treatment with CsA [34]. In retinal hypoxia after intraocular pressure elevation, CsA also proved to be a potent neuroprotective agent [25]. In this work, concentrations of 1, 3, 6, and 9 µg/mL were chosen since higher concentrations induced a decrease in cell viability in a previously published study using RGC-5 cells [26]. In a study by Kaminska et al., slightly higher concentrations (8–40 µM) led to the apoptosis of mixed neuronal/glial cultures, where TUNEL^+^ staining was observed only in neurons while astrocytes were unaffected [35]. For the biocompatibility testing performed in the present study, different concentrations of CsA were used without hypoxia, and we investigated the influence on the number of RGCs, macroglia, and microglia. The amount of RGCs did not change compared the control numbers. No evidence of a deleterious effect of CsA was found with double staining for TUNEL^+^ and Brn3a^+^ cells (Figure 1). Further in vivo experiments are necessary in order to exclude the long-term toxicity of CsA. However, in the course of our experiments and during the short observation period, no direct toxicity on RGCs could be detected. In accordance, no relevant micro- or macroglial activation was found with CsA treatment, although it can be mentioned that a missing macroglial response was observed in porcine ex vivo cultures [29,36,37].

In our retinal hypoxia model, hypoxia induced a reduction in retinal thickness, which is in line with our previously reported results [30,31]. CsA had a dose-dependent protective effect in retinal explants against reduction in retinal thickness as measured by OCT (Figure 3). The percentage of RGCs inside the ganglion cell layer was ameliorated after hypoxic stress by 9 µg/mL CsA (Figure 3). The other concentrations (6 µg/mL CsA or lower) did not show a rescue effect, although 6 µg/mL CsA reduced the percentage of TUNEL^+^ RGCs. Although only a macroglial response (GFAP staining) was observed after hypoxic treatment and was even strengthened by 6 µg/mL CsA treatment, the effect of hypoxia was completely counteracted with 9 µg/mL CsA (Figure 4). These results match the findings of a study investigating the effects of cyclosporine A treatment on brain remodeling after stroke in rats. CsA was shown to significantly reduce astrogliosis [38].

On a functional level, only 9 µg/mL CsA counteracted the complete loss in spontaneous RGC activity after retinal hypoxia (Figure 6). The discrepancy between functional and morphological results, especially the complete loss of RGC activity in the hypoxia group and the 6 µg/mL CsA + hypoxia treatment group, was already reported in one of our previous publications [39]. In this work almost no RGC activity was detectable after a hypoxic insult, although the mRNA expression of the RGC marker *Tubb3* was substantially retained. One possible explanation might be that the hypoxic insult left the RGCs structurally intact but functionally inactive. The same effect has been seen in neurons in the penumbra after ischemic stroke [40,41]. Two main molecular pathways seem to be important, by which CsA induces its neuroprotective effect. First of all, CsA inhibits the mitochondrial permeability transition pore (mPTP) [18,21], which if opened induces caspase-mediated apoptosis, and secondly induces calpain calcineurin cleavage [21,42,43]. Both pathways, if activated, trigger cell death independently from the immune system. Retinal ischemic injury upregulates cyclophilin-related protein D (CypD) protein expression, which is an essential structural component of the mPTP [25,44]. Moreover, inhibition of CypD by CsA treatment promotes RGC survival and blocks cell death in ischemic retinal injury [25]. Additionally, CsA has been shown to induce a dose-dependent reduction in the expression and activity of caspase 3 [25,26]. Results consistent with the work of Kim et al. regarding the survival of RGCs and activation of macroglia were noted. This is despite the fact that, in the work of Kim et al., they administered CsA systemically in their animal model differed from our local application in the culture medium of retinal organ cultures [25]. In a potential clinical situation, local treatment would be applied via intravitreal injection. In a chronic disease like glaucoma, a constant intravitreal application of neuroprotective CsA is probably not the most favorable type of application as every injection also bears the possibility of endophthalmitis. In vascular retinal diseases like diabetic retinopathy and vein occlusion, however, intravitreal application would be a suitable treatment method as anti-VEGF agents need to be injected anyway, and a local treatment is desired. 

As has already been reported, CsA reduces VEGF production, which would be a favorable side-effect [45]. VEGF is secreted by cells exposed to hypoxia and serves primarily to improve tissue vascularization to enhance its own oxygen and nutrient delivery [46]. VEGFs inhibited by CsA thus prevent new vessel formation, which is important for regeneration [47]. This inhibitory effect can also be assumed in the retina. To investigate this in the retina, further in vivo experiments are necessary. 

In summary, 9 µg/mL CsA presented good biocompatibility and induced a neuroprotective effect after hypoxic treatment. The observed neuroprotective effect is in line with the previously published data [25,26]. Thus, CsA might have additional therapeutic potential in the treatment of glaucoma or ischemic retinal vascular diseases like diabetic retinopathy or retinal vein occlusion. Therefore, in vivo animal studies treating an ischemic insult with intravitreal CsA injections are needed to confirm these results and to further evaluate the therapeutic potential of CsA as neuroprotective substance for retinal diseases.

## 4. Materials and Methods

### 4.1. Retinal Explant Preparation

Retinal explant preparation was performed as described previously [30,31]. Declaration according to §4.3 Animal Protection Act from 01.07.2016. Briefly, P14 to P15 Lister Hooded rats (Charles River, Sulzfeld, Germany) were euthanized with CO_2_ inhalation, enucleated, and the retinal explant was prepared under a sterile hood in cold R16 basal medium (Thermo Fisher Scientific, Needham, Waltham, MA, USA). Retinal explants were transferred to culture plate inserts (Corning, New York, NY, USA) with the retinal ganglion cell layer facing upwards and were cultured at 37 °C, 5% CO_2_, and 95% relative humidity atmosphere. 

### 4.2. Hypoxia

To induce severe retinal hypoxia ex vivo, we used our previously published retinal hypoxia organ culture model [30,31]. After placing six-well plates with retinas into our hypoxia chamber, the retinas were kept in an almost pure nitrogen atmosphere for 75 min with a pressure of one bar. To ensure that the residual air escaped from the chamber, the opposite opening was kept open during the first 5 min. Subsequently, the opening was plugged to maintain the hypoxic environment for the next 70 min. 

### 4.3. Cyclosporine A Treatment

CsA was added in different concentrations to the R16-complete media [30]. The concentrations used were 1, 3, 6, and 9 μg/mL. After 24 h of incubation, the medium was exchanged to R16-complete medium (without CsA). In the samples treated with hypoxia, CsA was added after the treatment.

### 4.4. Optical Coherence Tomography (OCT)

Optical coherence tomography scans and infrared measurements were performed as described previously [48]. Briefly, retinal explants were scanned by a Spectral-Domain OCT (Heidelberg Engineering, Heidelberg, Germany) after 24 and 48 h of cultivation. To perform scans with identical orientation of the same retinal explants, the position of the membrane insert was marked at the first measurement, and infrared images were taken shortly before the OCT scans to ensure identical positioning of the insert in both of the consecutive measurements. Volume scans were conducted at five equally distributed measuring points and analyzed by two blinded investigators.

### 4.5. Immunohistochemistry

For fixation, 4% PFA was pipetted directly onto the explants for 30 min. PFA was removed and retinal explants were washed with PBS for 5 min. Next, first 15% and then 30% sucrose solution was added to the retinas for 15 min each. Afterwards the explants were frozen in liquid nitrogen. Retinas were cut on a cryostat (12 μm sections). TUNEL staining (TdT-mediated dUTP-biotin nick-end labeling) was performed according to the manufacturer’s protocol using the In Situ Cell Death Detection Kit, Fluorescein (Roche, Basel, Switzerland) to visualize cell death. DAPI (4′6′-diamidino-2-phenylindole; Sigma-Aldrich, Taufkirchen, Germany) and Brn3a (primary antibody: Brn3a antibody SC-31984, Santa Cruz, Dallas, TX, USA; secondary antibody: Cy3 rabbit anti-goat (#305-167-003, Dianova GmbH, Hamburg, Germany)) staining was performed as described previously [30]. Photographs were taken using an Axioplan 2 imaging fluorescence microscope (Zeiss, Oberkochen, Germany) and analyzed by a masked investigator. For evaluation, all cells labelled with Brn3a antibody (retinal ganglion cells) of the GCL were counted in a blinded manner. Ten to twelve pictures were taken per each sample (*n* = 6) of every group for the quantification. The amount of RGCs was set in relation to the total cell amount (DAPI^+^ cells) of the GCL. In addition, TUNEL^+^ and Brn3a^+^ cells were counted as apoptotic RGCs. The TUNEL^+^ RGCs were compared with the total number of RGCs. 

For glial staining with GFAP (Z0334, Agilent, Waldbronn, Germany) and CD11b (#MCA275R, Bio-Rad (Abd Serotec-), Feldkirchen, Germany) sections were treated in the same way, but without TUNEL staining. Signal areas were evaluated with ImageJ software, as described previously [49]. Briefly, pictures were masked and converted into greyscale pictures, the background was subtracted, and the threshold was determined. An ImageJ macro was then used to calculate the signal area fraction.

### 4.6. Microelectrode Array Recording

Microelectrode arrays (MEAs) were incubated for 20 min with poly-L-lysine hydrobromide (Sigma-Aldrich, Taufkirchen, Germany) and washed three times with Ames’ medium (Sigma-Aldrich, Taufkirchen, Germany). After 48 h of cultivation, retinal explants were mounted on MEAs with the retinal ganglion cell layer facing to the array. The MEAs contain 60 TiN electrodes (8 × 8 grid, 30 µm electrode diameter, 200 µm electrode spacing, Multi Channel Systems, Reutlingen, Germany). One milliliter of warm (35–37 °C) oxygenated Ames’ medium was added, and the MEA was placed into an MEA amplifier (Multi Channel Systems, Reutlingen, Germany). Afterwards, the retinas were allowed to adjust to the new medium for 10 min, then the spontaneous RGC activity was recorded for 5 min (0.3–3 kHz filter; threshold for spike detection: six times electrode noise standard deviation). The number of electrodes that recorded spikes was analyzed as previously described [39]. 

### 4.7. Quantitative Real-Time PCR

mRNA was isolated from frozen punches and reverse transcribed using the MultiMACS cDNA Synthesis Kit (Miltenyi Biotec, Bergisch Gladbach, Germany) on the MultiMACS™ M96 Separator (Miltenyi Biotec, Bergisch Gladbach, Germany) according to the manufacturer’s protocol. After cDNA synthesis, quantitative real-time PCR was performed with 40 cycles using the SYBR Green SsoAdvanced Universal SYBR R Mastermix (Bio-Rad Laboratories, Feldkirchen, Germany) on a thermocycler (CFX96 Touch Real-Time PCR Detection System, Bio-Rad Laboratories, Feldkirchen, Germany). The cDNA expression levels of *Tubb3* was normalized to the cDNA level of the housekeeping gene *GAPDH*. The primers were designed using the Primer3 software, based on the published GenBank sequences (GenBank: KM035791.1, http://www.bioinformatics.nl/cgi-bin/primer3plus/primer3plus.cgi (accessed on 10 October 2017)). The final concentration of primers was 100 nM each. For the reaction mixture, 1 ng/µL of cDNA was used in a reaction volume of 20 µL. Each sample was analyzed in duplex. The relative expression of the target gene in the treated and control groups was calculated with the 2-^ΔΔ^ Ct method [50] and expressed as the fold changes in gene expression. The housekeeping gene encoding GAPDH served as endogenous control (*Tubb3* for TGAGGCCTCCTCTCACAAGT; *Tubb3* rev TGCAGGCAGTCACAATTCTC; *GAPDH* for GGCATTGCTCTCAATGACAA, *GAPDH* rev TGTGAGGGAGATGCTCAGTG).

### 4.8. Statistical Analysis

There was a Gaussian distribution of the residuals and an equal distribution of the standard deviation. Therefore, the ANOVA test followed by post-hoc analysis with the Tukey test was used to analyze group differences with respect to the control, CsA, or hypoxia treatment groups. Data are presented as mean +/− SEM. For all statistical tests, significance is indicated using the following significance levels: * *p* < 0.05, ** *p* < 0.01, and *** *p <* 0.001.

## Figures and Tables

**Figure 1 ijms-22-10196-f001:**
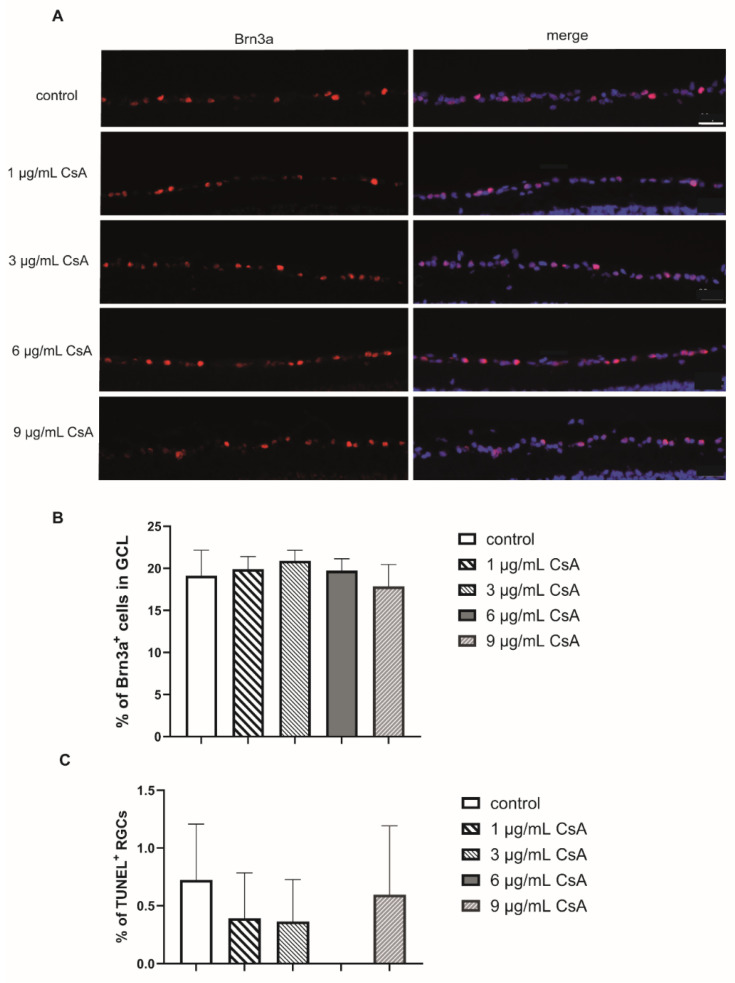
Biocompatibility of the different CsA concentrations. (**A**) Representative pictures of the Brn3a in each group after 48 h. Scale bar indicates 50 µm. (**B**) Quantification of Brn3a^+^ retinal ganglion cells (RGCs) labeld in red and cell nuclei (DAPI) in puple. The bar graph represents the percentage of these cells in the ganglion cell layer (GCL). (**C**) The percentage of TUNEL^+^ RGCs after 48 h of cultivation. Sections were double stained for DAPI and Brn3a. Bars and error bars indicate mean + SEM. (controls *n* = 10; all CsA groups *n* = 4–7).

**Figure 2 ijms-22-10196-f002:**
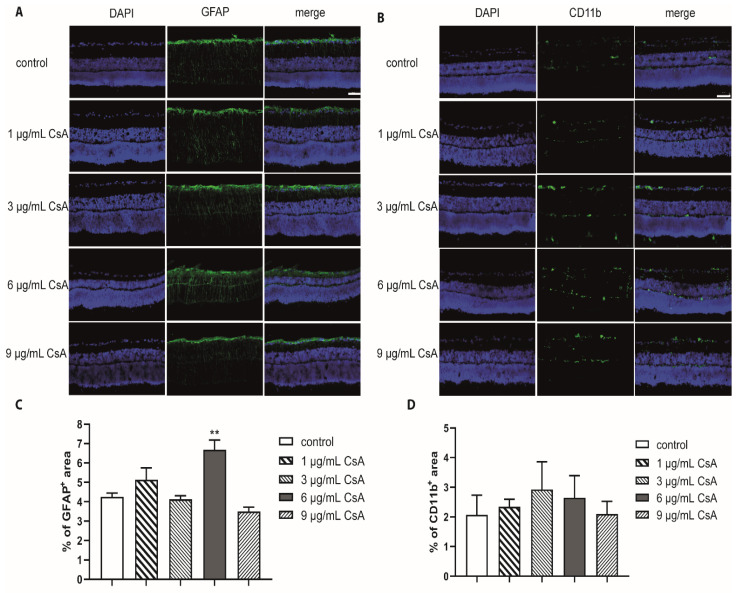
Influence of CsA on macro- and microglia. (**A**) Representative GFAP staining (green) and cell nuclei (purple) of retinas cultured for 48 h and treated with CsA at different concentrations (1 μg/mL, 3 μg/mL, 6 μg/mL, 9 μg/mL). (**B**) CD11b staining (green) of retinas after 48 h in culture and treatment with CsA at different concentrations. Cross sections of the retinas are shown at 200× magnification. The scale bar indicates 50 µm. (**C**) The values in the graph present the percentage of the GFAP-stained area compared to the total area of the retina. The evaluation was performed after 48 h. The percentage of macroglia remained almost unaffected at the 1 μg/mL, 3 μg/mL and 9 μg/mL concentrations compared to controls. At the concentration of 6 μg/mL, the level increased significantly compared with controls (*p* < 0.01). (**D**) The percentage of microglia did not change compared to the controls. Data are depicted as mean + SEM, with ** *p* < 0.01 versus control (controls *n* = 10; all CsA groups *n* = 4–7).

**Figure 3 ijms-22-10196-f003:**
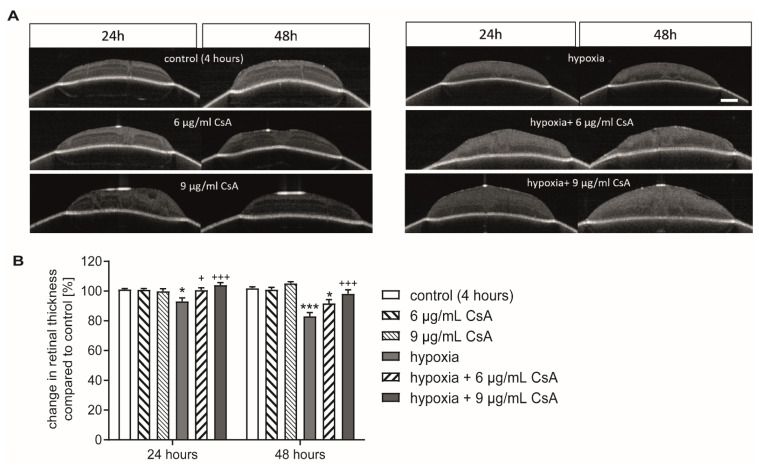
CsA protected against loss in retinal thickness. (**A**) Representative pictures of the OCT scans after 24 and 48 h. With the first measurement the position of the membrane insert was marked, and infrared images were taken shortly before the OCT scans to ensure identical positioning of the insert in the two consecutive measurements. Retinal thickness was measured at five equally distributed measuring points. Scale bar = 200 µm. (**B**) Bar graph represents the change in retinal thickness at the 24 h or 48 h time points compared to the measurement at the 4 h time point, which is set as control. Bars and error bars indicate mean + SEM, *n* = 5 for all treatment groups. Significance is indicated as * with respect to the control condition and as + with respect to the ischemic treatment using the following significance levels: +/* *p* < 0.05, +++/*** *p* < 0.001.

**Figure 4 ijms-22-10196-f004:**
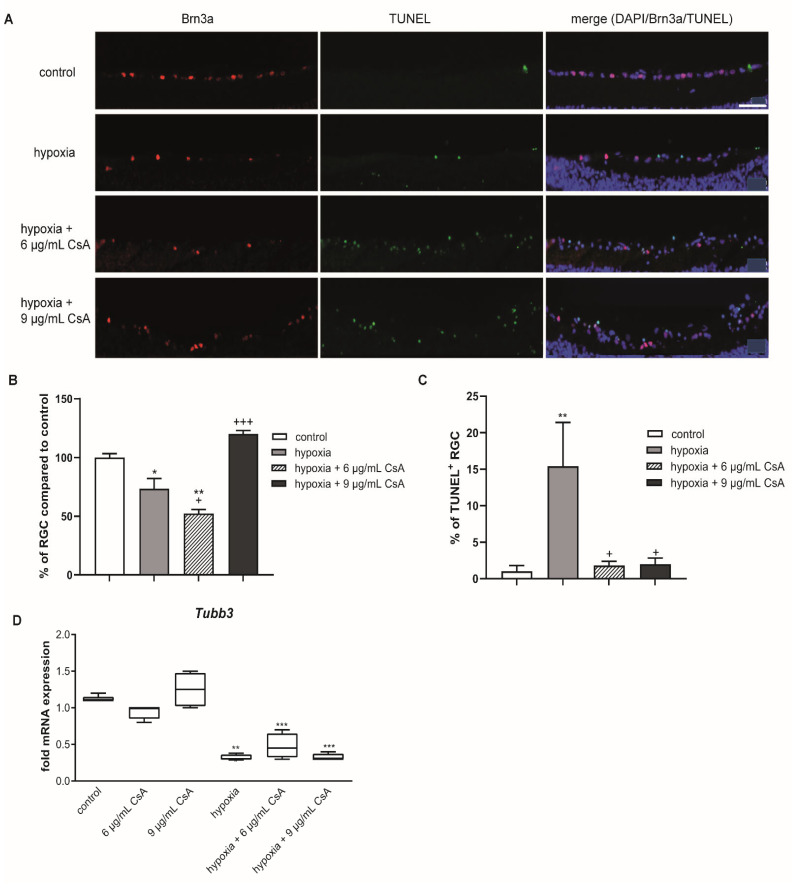
CsA treatment rescued RGCs from hypoxia-induced apoptosis. (**A**) Representative pictures of the corresponding sections. Sections were triple stained for DAPI (purple), Brn3a^+^ (red) and TUNEL^+^ (green) cells. Scale bar indicates 50 µm. (**B**) Bar graphs represent the percentage of RGCs in the GCL after 48 h of cultivation. Hypoxia caused a loss of RGCs, which could be counteracted by treatment with 9 µg/mL CsA. (**C**) Graphical illustration of TUNEL^+^ RGCs out of the total RGC amount as a percentage. Hypoxia treatment led to a massive increase of TUNEL^+^ RGCs, which was prevented by treatment with 6 or 9 µg/mL CsA. (**D**) Retinal hypoxia resulted in a significant reduction of neuronal mRNA expression of the RGC marker *Tubb3*. CsA treatment with either 6 or 9 μg/mL could not protect from this loss in neuronal mRNA expression after hypoxia. CsA did not alter neuronal *Tubb3* mRNA expression in untreated retinal explants. Data are depicted as mean + SEM. Significances are indicated as * with respect to the control condition and as + with respect to the ischemic treatment using the following significance levels: +/* *p* < 0.05, ** *p* < 0.01, +++/*** *p* < 0.001.

**Figure 5 ijms-22-10196-f005:**
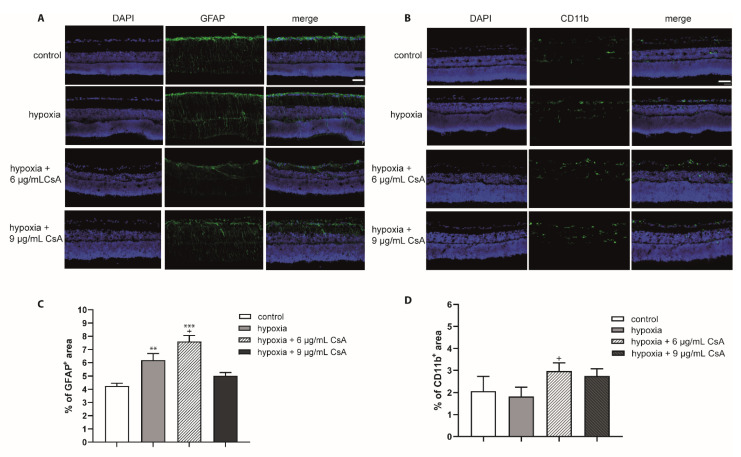
Effects of hypoxia and CsA treatment on glial cells. Immunohistochemical staining (GFAP and CD11b) of hypoxia- and CsA-treated retinas: representative staining of retinas cultured for 48 h. (**A**) Macroglia were labeled using a GFAP antibody (green) and cell nuclei labled with DAPI (purple); (**B**) microglia were stained with a CD11b antibody (green) and cell nuclei labeled with DAPI (purple). (**C**) The percentage of the GFAP^+^ area compared to the total area is presented in the diagram. Scale bar indicates 30 µm. The percentage of the macroglia area was higher than that of pure hypoxia at 6 µg/mL CsA. The higher concentration of 9 μg/mL CsA maintained GFAP abundance almost at the level of control retinas. (**D**) The percentage of microglia was higher in CsA-treated retinas (approximately 2.5% to 3.25%) than in controls (approximately 2%) and pure hypoxia (approximately 1.8%). Significances are indicated as * with respect to the control condition and as + with respect to hypoxia treatment using the following significance levels: ^+^ *p* < 0.05, ** *p* < 0.01, and *** *p* < 0.001.

**Figure 6 ijms-22-10196-f006:**
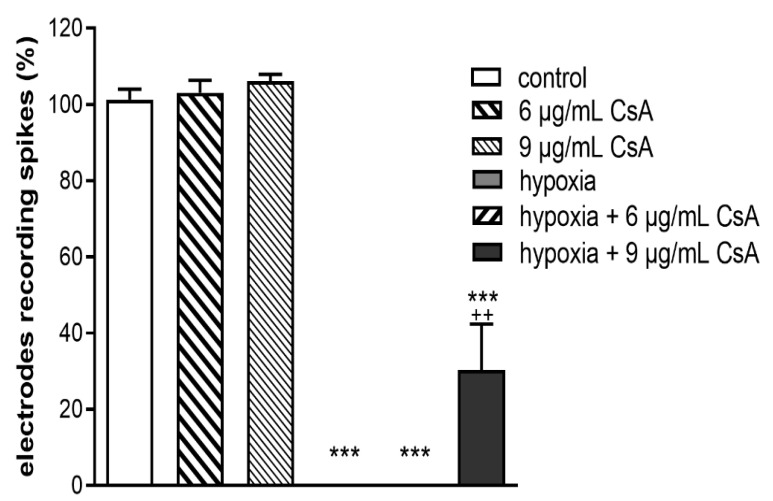
Spontaneous RGC activity. Bar graph represents the spontaneous RGC activity measured via multi-electrode assays (MEAs) with the control set to 100%. Hypoxia affected the spontaneous RGC activity, and this effect could not be reversed with treatment of 6 µg/mL CsA. However, the loss of function after retinal hypoxia was counteracted up to 30% activity, but not reversed by 9 µg/mL CsA (*p* < 0.001). Bars and error bars indicate mean + SEM, *n* = 3–5 for all treatment groups. Significances are indicated as ^+^ with respect to the control condition and as * with respect to ischemic treatment using the following significance levels: ^++^ *p* < 0.01, *** *p* < 0.001.

## Data Availability

The data presented in this study are available in the article.

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
