# Peer review of "Cyclosporine A Protects Retinal Explants against Hypoxia"

_ijms, 2021, doi:10.3390/ijms221910196_

Round 1

Reviewer 1 Report

The authors have significantly improved the manuscript and incorporated all the suggestions. 

Only minor corrections need the authors attention:

  • Delete sentence lines 44-45 as the same sentence is at the end of the paragraph (line 48-49)
  • Please correct the sentence in line 92
  • The authors should incorporate lines 100-109 into section 2.4 or deleted them as they are essentially repeated in the section "glial cell response"
  • References 29 and 38 correspond to the same article but different years. 

Author Response

The authors have significantly improved the manuscript and incorporated all the suggestions. 

Only minor corrections need the authors attention:

  • Delete sentence lines 44-45 as the same sentence is at the end of the paragraph (line 48-49)
  • Please correct the sentence in line 92
  • References 29 and 38 correspond to the same article but different years. 

Response: We thank the reviewer for the assessment and careful examination of the manuscript. All suggestions were implemented.

  • The authors should incorporate lines 100-109 into section 2.4 or deleted them as they are essentially repeated in the section "glial cell response"

Response: We understand the point of view but in this chapter, we will examine possible direct or indirect side effects of CsA on glial cells. In chapter 2.4, the influence of hypoxia and CsA treatment was investigated. We have reworded the part somewhat

“The immunosuppressive drug CsA may affect inflammation and the wound healing process. It is also possible that there could be indirect effects of CsA on glial cells via other cell types. Therefore, the influence of CsA treatment on macro- and microglial cells was also investigated by staining the markers glial fibrillary acidic protein (GFAP) and cluster of differentiation molecule 11B (CD11b).

Reviewer 2 Report

Authors have done experiments using retinal explant model mostly concentrating to the retinal ganglio cell responses upon hypoxic conditions and testing protective effects of cyclosporin A upon hypoxia. Some minor language checking is still needed. Nonetheless, authors have done a lot of improvements to the manuscript. Picture are visible and clearly marked. There is added a lot of text to improve manuscript throughout.

Introduction

- lines 43-49, Please modify. Some part of the text appear as double.

- lines 62-64, ”CsA was administered by intraperitoneal injections in 4-month-old C57BL/6 mice once a day with a concentration of 5 mg/kg (corresponding approximately to a total quantity of 100 µg)…”

Underlined text not sound very scientifical and is not needed. If dose for mice was 5 mg/kg, it tells enough to reader. It is not needed to tell the amount because it depend on the weight of each mouse and based to that. Underlined text is estimated value but not exact and not needed to report.

In addition, please clarify whole sentence. It not make sence and is not understandable. Example divide into two sentences. It would make it clearer.

- lines 81-82, please, clarify little bit the end of the sentence.

Results

- lines 92-93, please clarify. Hard to understand results interprut.

- lines 97-98, ”Concerning the morphologic RGCs analysis, no changes in size and shape of RGCs were detectable.”

Please, clarify how this was detected and for which result conclusion based on. If its just common attention maybe not good to present. Needs results to support interpretation. Explain little bit more detail, cell pictures, results?

- line 106, correct text

- Please bring figures near result presented. Just after spesific title and text. Not all at the end of whole result section.

- Results 2.1. says that ” Analysis of the percentage of TUNEL+ RGCs did not reveal significant changes in any of the CsA concentrations at any of the two investigated time points compared to the controls (Fig 1C).”       How about CsA 6 μg/ml?

- Figure legend 2, ” D) The percentage of microglia increased considerably at concentrations of 1, 3, and 6 μg/mL compared with controls. At the 9 μg/mL concentration, the percentage was almost equal to the controls.”

Please not interpret result like that if there is not significant changes as there is not marked. If there is that is ok, but otherwise can not say that there was effects. Each interpretation needs to basicly based to the statistical analyses. There is changes or not, no any verbal changes without statistic.

- Please refer/cite to the figure 2 earlier that it is more easy to follow text and exact result figure. Refer/cite to the figure 2 example at the line 104.

- Please, open as well result of Figure 2B in the text e.g. after GFAP results from line 109. CD11b result text in the figure 2 legend not sound good.

- line 126, please correct or clarify sentence.

-line 169, please clarify sentence.

Discussion

-lines 283-286, make sentence clearer and shorter e.g. few different sentences.

-line 293, VEGF is opened but it already appear few line above (anti-VEGF). VEGF needs to open already there.

 Material and methods

- lines 356-367, Use [  ] insed of ( ). Do not use as double (    )). Same in line 366.

-line 409, correct ANOVA

Author Response

Authors have done experiments using retinal explant model mostly concentrating to the retinal ganglio cell responses upon hypoxic conditions and testing protective effects of cyclosporin A upon hypoxia. Some minor language checking is still needed. Nonetheless, authors have done a lot of improvements to the manuscript. Picture are visible and clearly marked. There is added a lot of text to improve manuscript throughout.

Introduction

- lines 43-49, Please modify. Some part of the text appear as double.

Response: We thank the reviewer for the hint. The double sentences was removed.

- lines 62-64, ”CsA was administered by intraperitoneal injections in 4-month-old C57BL/6 mice once a day with a concentration of 5 mg/kg (corresponding approximately to a total quantity of 100 µg)…”

Underlined text not sound very scientifical and is not needed. If dose for mice was 5 mg/kg, it tells enough to reader. It is not needed to tell the amount because it depend on the weight of each mouse and based to that. Underlined text is estimated value but not exact and not needed to report.

In addition, please clarify whole sentence. It not make sence and is not understandable. Example divide into two sentences. It would make it clearer.

Response: Thank you for that valuable hint. We have deleted the underlined text and clarified the unclear sentence in two sentences.

“CsA was administered in 4-month-old C57BL/6 mice once a day with a concentration of 5 mg/kg. Both CsA and vehicle control were injected once by intraperitoneal injection at 24 h before and at the time of initial IOP elevation.[25].“

- lines 81-82, please, clarify little bit the end of the sentence.

Response: We have supplemented the sentence:

“These findings further support CsA as a potential therapeutic agent for diseases affecting RGC-like glaucoma or central artery occlusion, because CsA could possibly prevent RGC loss and mitigate a harmful immune response.”

Results

- lines 92-93, please clarify. Hard to understand results interprut.

Response: We thank the reviewer for this hint, we corrected the sentence.

“In detail, quantification of RGCs after 48 h of incubation revealed 19.1% Brn3a+ cells in the GCL in the control group.”

- lines 97-98, ”Concerning the morphologic RGCs analysis, no changes in size and shape of RGCs were detectable.”

Please, clarify how this was detected and for which result conclusion based on. If its just common attention maybe not good to present. Needs results to support interpretation. Explain little bit more detail, cell pictures, results?

Response: We agree with the reviewer and deleted this part of the sentence.

- line 106, correct text

Response: Thank you for the advice, the text was corrected.

To evaluate gliosis, the percentage of GFAP+ or CD11b+ fluorescence area was measured, and the change compared with the control group was assessed.”

- Please bring figures near result presented. Just after spesific title and text. Not all at the end of whole result section.

Response: I'm sorry it's inconvenient for the reading flow. Unfortunately, the format template is designed so that the Figures should appear in Section 2.6-at least that's how I understand it

- Results 2.1. says that ” Analysis of the percentage of TUNEL+ RGCs did not reveal significant changes in any of the CsA concentrations at any of the two investigated time points compared to the controls (Fig 1C).”       How about CsA 6 μg/ml?

Response: Please Excuse the unclear wording in the 6µg/mL group no apoptotic cells were counted. We have now added this to the text.

- Figure legend 2, ” D) The percentage of microglia increased considerably at concentrations of 1, 3, and 6 μg/mL compared with controls. At the 9 μg/mL concentration, the percentage was almost equal to the controls.”

Please not interpret result like that if there is not significant changes as there is not marked. If there is that is ok, but otherwise can not say that there was effects. Each interpretation needs to basicly based to the statistical analyses. There is changes or not, no any verbal changes without statistic.

Response: We thank the reviewer for this hint and change the figure legend accordingly:

“The percentage of microglia did not change compared to the controls.”

- Please refer/cite to the figure 2 earlier that it is more easy to follow text and exact result figure. Refer/cite to the figure 2 example at the line 104.

Response: We thank the reviewer for this hint and cite Figure 2 at line 108 (former 104)

- Please, open as well result of Figure 2B in the text e.g. after GFAP results from line 109. CD11b result text in the figure 2 legend not sound good.

Response: We totally agree with the reviewer and changed it accordingly

- line 126, please correct or clarify sentence.

Response: We thank the reviewer for this hint and corrected the sentence:

“At a concentration of 9 µg/mL, CsA protected the retinal explants from thinning at both time points (24 h: 104.6%, p<0.001; 48 h: 98.2%, p<0.001) and ameliorated the hypoxic effect completely (Fig. 3 B).”

-line 169, please clarify sentence.

Response: Many thanks for the hint, we totally oversee the wrong wording.

Discussion

-lines 283-286, make sentence clearer and shorter e.g. few different sentences.

Response: We thank the reviewer for this advice and modified the sentence:

Consistent results with the work of Kim et al. regarding the survival of RGCs and activation of macroglia were noted. This is despite the fact that the work of Kim et al. in which they administered CsA systemically in their animal model differed from our local application in the culture medium of retinal organ cultures”

-line 293, VEGF is opened but it already appear few line above (anti-VEGF). VEGF needs to open already there.

Response: Thank you for that hint. It was corrected in the text.

 Material and methods

- lines 356-367, Use [  ] insed of ( ). Do not use as double (    )). Same in line 366.

Response: Thank you for this hint, we changed it accordingly

-line 409, correct ANOVA

Response: We thank the reviewer for this correction!

This manuscript is a resubmission of an earlier submission. The following is a list of the peer review reports and author responses from that submission.

Round 1

Reviewer 1 Report

Schultheiss et al., assessed the neuroprotective role of Cyclosporine A on the ganglion cell survival in the retinal explants against hypoxia. While the scope of the study is of interest and presents a subsequent findings in comparison to the previous study of the group, manuscript is written in a way that it is extremely difficult to understand the findings. The authors must significantly improve the text of the manuscript. In the current form, the information is stated without flow of thoughts or the background to facilitate understanding the subject. There are far too many language issues and inconsistencies in the data presentation. Authors should significantly improve the data presentation before any scientific issues could be further addressed. 

Here are just some of the examples of the issues in the manuscript:

In the Introduction, lines 55-57, the authors provide the exact dosing of a previously published work without giving any explanation of how this info is relevant. Line 60, authors refer to the “RGC-5 issue”. What is the RGC-5 issue? Line 65, authors claim that CsA would be trapped in the vitreous. Why is it so? Is there any previous work to support this? Line 68, what does the local half-life means?

In the result section, it is not clear what was the research question in different experiments. Authors should clarify that the experiments shown in Figure 1 were used to determine if the CsA treatment may have potential side effects in the healthy retina. 
In line 74 authors say that they quantified the total amount of cells and the number of RGS in GCL. They only show quantification of RGC cells. I wonder if they meant that they quantified DAPI+ nuclei in GCL and Brn3a+ cells in GCL? In figure 1 in the magnification of the RGC does not allow for any determination of the cell morphology. 
Why did authors mention the investigation of inflammatory status in the retina if none of the data are shown? Why are data not shown?

Why in OCT images in Figure 2 different layers in the retina are not distinguishable in the case of hypoxia and hypoxia+CsA? For these experiments was the same explant used first as in the hypoxia treatment and then treated with CsA? If not, how did the authors control if the retinal thickness changes are not due to the different explants from different animals? 
Examples of the retinal cross-section shown in figure 3A do not represent the quantification. There are only 2 TUNEL+ cells in hypoxia, while in CsA treatments, there are much more + cells. Again this magnification doesn’t allow any detailed view in the RGC morphology. 

There are quite a few additional issues that I hope will be solved once the authors improve the data presentation.

Reviewer 2 Report

General

In the present study, Schultheiss et al investigated if Cyclosporine A protects retinal explants against hypoxia. Explant was dissected from eye of rats and cultivated. Cyclosporine A was then administrated as ex vivo. Different methods were used to evaluate the potential of cyclosporin A to improve of hypoxic condition. There was found with highest concentration has positive effects and in discussion section was evalauted the potential to move into in vivo studies and probably as aim of clinical use of in vitro injection.

in general, manuscript needs a lot of improvement. Scientific language needs to modify in many parts and need to avoid too much extra explanation. Time forms need to check as scientifically correct. Few results need to ensure be correct (Figure 4A and C, Figure 3A and C).

Figure legends need to clarify and make more compact. Figures need to move closer spesific results part in text, rigth after mentioned in the text. Not include all pictures into one section. Add figures to the section mentioned in the text. Move results explanations from legends that should appear in text near/before of the figure.

Check language and that all abbreviations ar opened when mentioned first time.

Title + first page

-Add cities for informed affiliations. Check fonts for authors and affiliation informations. Add authors email and initials and other infromations for corresponding author (email, phone).

-Add affiliation number 5 for some author or delete it.

 Abstract

- line 18, there is marked CsA concentrations as (1-, 3-, 6-, or 9 μg/ml). It would be better mark used concentrations in abstract shortly as “1 μg/ml to 9 μg/ml”. Or if all concentrations is wanted to inform as “1 μg/ml, 3 μg/ml, 6 μg/ml, or 9 μg/ml”.

- Is the “Brn3a” established name? Open Brn3a when mentioned first time if its possible. Open both in abstract and in main text when mentioned first time.

-line 19, confusing style “..the percentage of (apoptotic) RGCs”. why word “apoptotic” is in parenthesis? It should be without parenthesis if this means that there is detected apopotic PGCs.

- Open abbreviations, RGC and OCT-scans, when mentioned first time both in abstract and main text.

-line 20, Sentence starting from line 20 is confusing. Please, clarify. Probably there is missing verb or right word order is missing . e.g. “OCT-scans were done/performed to measure the…” Please, make the sentence little bit clearer. Same in the middle of sentence “…and recordings with multielectrode array (MEA)” Modify the vebr to not be the end of the sentence.

Then it would be easier for reader to follow the text.

-line 22, Mention shortly the used method to induce hypoxic condition. It can be included into the sentence

 “To examine the neuroprotective, retinas were subjected to a hypoxic insult prior to CsA treatment”.

-  Lines 23-34 “The tested CsA concentrations had no effect on RGCs and microglia cells. A neuroprotective effect after hypoxic insult on RGCs was demonstrated at a concentration of 9 μg/ml CsA.”

Please, clarify. First there is metioned that there was not any effect and then introduced neuroprotective effects of 9 ug/ml. Please, clarify and give examples more detailed that reader can follow if there was not not effects.

-line 28, please start the sentence more scientifical way “Therefore, CsA should be tested in further in vivo experiments to determine whether local treatment is suitable for the treatment of hypoxic retinal diseases.”

example “There is needed further in vivo examinations to determine whether local CsA treatment is suitable option for hypoxic retinal diseases”

-It is little bit confusing at the beginning of the manuscript that has there used animals and dosing before investigations, refering to the lines 55-56 and line 61 “organotypic retinal explants…”.

Please clarify from the beginning, if organotypic retinal explants are originally collected from animals. Now, it comes clearly at the end of the manuscript but at the beginning there is few confusing parts related to that.

In abstract there is not mentioned of animals and suddenly in introduction, there was mentioned something about animals. Then in material section it was clear but need to mention also shortly in the abstract to avoid confusing.

Introduction

-lines 40-41, Please, move references at the end of the sentence “ Hypoxia plays a role in many ocular diseases such as glaucoma [2], diabetic retinopathy[3,4], age-related macular degeneration [5] (AMD), and central retinal artery occlusion (CRAO)[6].” Reader can pick up the spesificreference from the end of the sentence.  Also reference between abbreviation and explanation is not good way to express e.g. “age-related macular degeneration [5] (AMD)”

Changes helps text to flow.

-line 43, sentence “The second leading cause of irreversible blindness worldwide is glaucoma”. If you have not mentioned first leading cause, not mention second. Reader start to thinkin what is the first one causes of irreversible blindness. Please, clarify or modify sentences.

-line 50, Move references at the end of the sentence “ ischemic stroke[14-16] or traumatic brain injury[17-19].” Same in the next sentence.

- line 53, please clarify throughout the manuscript, if there is used mices and dosing before explant investigation. That should mention also in abstract. That arise in introduction part and remain unclear “Here, CsA was administered by intraperitoneal injections in 4-month-old C57BL/6 mice once a day with a concentration of 5mg/kg 24 h before and at the time of initial IOP elevation.” Clarify, that the situation is clear for reader from the beginning.

Now, it is confusing if CsA was first administrede for mices in the present study or does this refer for the previous sentence. Change word “here” more explanatory form. It is confusing, because in material section there is mentioned Rats. Please, clarify and take animals to introduction at the beginning from the abstract that reader is known the whole story of the results and have not to guess the situation.  

- line 57, Open abbreviation “IOP”

- lines 61 -63, Introduce someway and shortly mention findings in the previous study.

- line 69, Last section nicely introduced clinical aspect for the study and last sentence tight it to the present study. Still it need to little bit bound more to the present study. Continue last section to introduce the meaning of the present study for clinical perspective. Mention and highlight at the end of the main findings in the present study and introduce future perspective.

Results

 -Please, put results figures under the title it is introduced and near after first time mentioned/referred in the text. It is really hard to read text, if all results are later in one section.

 -lines 61-63, there is mentioned that “ The biocompatibility of 1-, 3-, 6- and 9 μg/ml CsA and the  neuroprotective potential was evaluated in our previously published retinal hypoxia  organ culture model”. Then results section start “Biocompatibility of Cyclosporine A” I

If that is already investigated why results are introduced in the present study. Please, modify in accordance with each other. And if there is reason to do it in both studies, mention also that.  Then reader not confusing with that.

-line 76, There is mentioned that “all investigated concentrations of CSA reduced the number of RGCs after 24 h of incubation”. But in the figure 1 (and also mentioned later on the text) there is shown that only one concentration significantly decreased RGC. Please clarify and be clear if there is significant decrease. If there is not significant reduction, introduce it with some other way/words exmaple “Other concentrations did not decrease significantly RCG even the levels were  xxx-fold of control”. “ 3 μg/ml CsA group significant reduced RCG levels, whereas the cell number was for Brn3a+ cells in the 1 μg/ml conc CsA as 22.36, 6 μg/ml as…”  As lines 80-81 have done for higher concentration. It is possible also move 1 μg/ml there and start to introduce with control and significant changes.

 It is better use exact term only in case in which the decrease/increase was significant. It is confusing that e.g. some concentration decreased something but also some other decreased it significantly.

-Check results section 2.1 and use scientific language again and make it more flow. Modify the existing text with mentioned tips.

-from line 81, Start again with controls and then introduce no significant changes. Now, it was confusing to see controls afterwards, text not flow and reader struggle with understanding. With small changes text will flow better.

- line 89, replace was to were

- from line 89, “Concerning the morphologic RGCs analysis all tested CSA concentrations showed good biocompatibility (Fig 1).” How you explain biocompatibility? Does it mean that CsA did not weakend cell viability and it was well tolerated?

-lines 88 – 94, Please not introduce results if not going to show data or remove it into discussion. Otherwise show data too. Again, open abbreviations. It could be done new section for results started from line 88, if its kept in the manuscript.

- from line 103, it is confusing that all concentration reduced thinning but only one is mentioned. Please, clarify and mention only significant reducing agent, but introduce all concentrations responses some other way.

-Result section 2.2 if controls are taken at timepoint 4 h it should mention in Figure 2. It is always better if you can add controls with same timepoints. Mention also in the text concentrations used in the measurement.

-line 108, add reference

- Figure 3, How highest concentration CsA improved situation in TUNEL test, is there more green dots than in hypoxia induced cells? Please, explain little bit immunohistochemistry results. in the picture looks like there is more TUNEL postivie cells in CsA treated cells than hypoxia or control cells. Also, DAPI staining is mentioned but not shown/pointed in Figure3A. Please, mention if it is included in last picture (Figure 3A). How massive increase of TUNEL positive cells with hypoxia in seen from figure 3A? Or does results (colums) based to exact measurements and not picture of 3A. Check results that correct.

- line 132, open GFAP when mentioned first time and line 144, open MEA

-lines 141-142, please clarify last sentence

- please, put figures in the order mentioned in the text (figure 5 and 4) first time.

- Figure 4, GFAP in pictures looks little bit different than in diagrams. Please, clarify and enlarge pictures. Check results that correct.

-line 148, make sentence clear, “ After hypoxic treatment though, no spontaneous RGC spiking was 148 detectable anymore”

- next two sentences from line 148, make clearer. Sentences are confusing. Was the responses with concentrations 6 μg/ml and 9 μg/ml same or different?  was both efficient, if does, put into same sentence not needed to repeat. If responses are different ,make that more clear for reader.

-All colourful figure pictures could be bigger to see better. (Figure 1A, 3A and 4A/B)

-Figure 3A à label error

 Discussion

 - There alter CsA and CSA, please modify throughout the manuscript and open if means different things.

-lines 231-247, explain a lot of results, and is more like results section. After that there is more like discussion. Reflect of results to the previous literature. Please modify little bit the beginning of discussion and add comparing previous findings between own results. Take away results that are not really needed to discuss.

-line 256, “Although our results are in accordance with the results of Kim et. al, the way how CsA was applied is very different”. Pleasy explain also how results are in accordance with Kim et al, because in the present study caspase-3 was not measured it need to open and explain little bit more.

Material and methods

-lines 291-294, please clarify, Cyclosporine A treatment explanation.

- complete reagent information, city, country

- correct text; lines 309-310

- How used statistical methods are justified? ANOVA and Tukey test?